# Comparing Prompt-Based and Standard Fine-Tuning for Urdu Text Classification

**Faizad Ullah**[1]**, Ubaid Azam**[1]**, Ali Faheem**[1]**, Faisal Kamiran**[2]**, and Asim Karim**[1]

[1]Computer Science, Lahore University of Management Sciences (LUMS), Lahore, Pakistan
[2]Computer Science, Information Technology University (ITU), Lahore, Pakistan
{faizad.ullah, 19030040, ali.faheem}@lums.edu.pk
faisal.kamiran@itu.edu.pk
akarim@lums.edu.pk

## Abstract

Recent advancements in natural language processing have demonstrated the efficacy of pre-trained language models for various downstream tasks through prompt-based fine-tuning. In contrast to standard fine-tuning, which relies solely on labeled examples, prompt-based fine-tuning combines a few labeled examples (few shot) with guidance through prompts tailored for the specific language and task. For low-resource languages, where labeled examples are limited, prompt-based fine-tuning appears to be a promising alternative. In this paper, we compare prompt-based and standard fine-tuning for the popular task of text classification in Urdu and Roman Urdu languages. We conduct experiments using five datasets, covering different domains, and pre-trained multilingual transformers. The results reveal that significant improvement of up to 13% in accuracy is achieved by prompt-based fine-tuning over standard fine-tuning approaches. This suggests the potential of prompt-based fine-tuning as a valuable approach for low-resource languages with limited labeled data.

## 1 Introduction

Recent advancements in natural language processing (NLP) have highlighted the efficacy of pre-trained language models (PLMs) in various downstream tasks, including text classification. PLMs, such as (Conneau et al., 2020; Devlin et al., 2019) and (Sanh et al., 2019), have revolutionized NLP by pre-training on extensive textual data to acquire language understanding and common knowledge. However, optimizing the performance of models for various languages, especially low-resource languages like Urdu and Roman Urdu, presents challenges that need to be addressed. Standard fine-tuning usually requires large amounts of task-specific labeled examples to adapt the parameters of PLMs for robust performance. More recently, prompt-based fine-tuning (Gao et al., 2021) has

emerged as a promising alternative for improving classification accuracy in low-resource language contexts (An, 2023; Lee et al., 2022; Jin et al., 2022; Schucher et al., 2022; Wang et al., 2022). Prompt-based fine-tuning combines a small set of annotated examples (few shot) with carefully designed language-specific prompts tailored to the task. These prompts explicitly guide the models, providing crucial context and information for precise predictions.

This paper presents an empirical evaluation of prompt-based fine-tuning with traditional standard fine-tuning for text classification in Urdu and Roman Urdu. The objective is to determine whether prompt-based fine-tuning surpasses the performance of standard fine-tuning of multilingual PLMs in few shot setting. To do this, we conduct experiments on five diverse datasets spanning different domains and encompassing various classification tasks. These datasets are carefully selected to represent the challenges and nuances of Urdu and Roman Urdu text classification. Additionally, we utilize three pre-trained multilingual transformers (1) BERT-Multilingual (Devlin et al., 2019), (2) DistilBERT (Sanh et al., 2019), and (3) XLM-RoBERTa (Conneau et al., 2020). These transformers have been widely used and proven effective in various NLP tasks across multiple languages. By incorporating numerous transformers, we aim to evaluate the robustness and generalizability of prompt-based fine-tuning across different architectures and language models.

Our findings reveal a significant improvement of up to 13% in classification accuracy achieved by prompt-based fine-tuning over traditional approaches. This improvement highlights the effectiveness of prompt-based methods in capturing the complex linguistic characteristics and nuances inherent in Urdu texts and demonstrates its potential as a promising alternative for low-resource languages where limited labeled data is available. To

the best of our knowledge, there are no published works on prompt engineering or prompt-based fine-tuning for Urdu text classification. Our work comparing prompt and standard fine-tuning in Urdu can seed further research in this direction.

## 2 Literature Review

In the past few years, social media platforms have experienced an enormous surge in users. With the rise in digital media usage, there is an increasing demand for automated text classification in Urdu.

Recently, various transfer learning and data augmentation approaches have been investigated for text classification, such as those described by (Banerjee et al., 2019; Azam et al., 2022; González-Carvajal and Garrido-Merchán, 2020; Alam et al., 2023). These methods use pre-trained language models and fine-tune them on smaller datasets to enhance their performance on specific tasks. However, a recent technique in natural language processing called prompt engineering has recently gained attention from researchers (Liu et al., 2023; Gao et al., 2021). Designing prompts to guide language models improves model predictions by effectively utilizing contextual information. This technique achieves better results with minimal data, including short learning or zero-shot learning. Various types of prompts are used for text classification, namely Human Designed Prompts (Brown et al., 2020), Schema Prompts (Zhong et al., 2022), and Null Prompts (Logan IV et al., 2022).

While prompt engineering research has primarily focused on the English language, there has been recent work exploring its effectiveness in other languages. For example, (Song et al., 2022) conducted research to determine if prompt engineering could improve text classification in Chinese. Their results showed that the use of prompts yielded positive results. Similarly, (Seo et al., 2022) applied prompt engineering techniques to Korean and found that it improved performance for various text classification tasks, including topic and semantic classification, even with few-shot learning. However, no known work has been done to study prompt learning for the Urdu language.

## 3 Experimental Design

In this section, we describe the datasets and the experimental setup employed to compare the performance of prompt-based and standard fine-tuning approaches for text classification in Urdu. Fine-tuning models on limited amounts of labeled data can introduce instability in execution and result in substantial performance variations depending on the choice of data splits (Zhang et al., 2021; Dodge et al., 2020). To generate robust results, we adopt a careful and comprehensive approach, as outlined below.

We aim to fine-tune a pre-trained language model $\mathcal{L}$ (standard and prompt-based) on task $D$ with label space $\mathcal{Y}$ for Urdu and Roman Urdu text classification. Our goal is to develop effective learning strategies that generalize well to an unseen test set $(x_{\text{in}}^{test}, y^{test}) \sim D_{\text{test}}$. In the few-shot setting, we have limited training examples per class. Let $K$ denote the number of training examples per class, and $|Y|$ denote the total number of classes in the task. Thus, the few-shot training set $D_{\text{train}}$ consists of $K_{\text{tot}} = K \times |Y|$ examples, where $D_{\text{train}} = \{(x_{\text{train}}^{(i)}, y_{\text{train}}^{(i)})\}_{i=1}^{K_{\text{tot}}}$. We utilize a development set $D_{\text{dev}}$ to select the optimal model and tune hyper-parameters. The size of $D_{\text{dev}}$ set is equal to the few-shot training set, i.e., $|D_{\text{dev}}| = |D_{\text{train}}|$.

Now, given the language model $\mathcal{L}$, our process begins by converting the input $x_{\text{in}}$ into a token sequence $\tilde{x}$, which is then mapped to a sequence of hidden vectors $h_k \in \mathbb{R}^d$ by the language model $\mathcal{L}$. For example, in a binary sentiment classification task, we can construct a prompt using the input $x = $ Yeh jaga bohat pyari hai. The prompt formulation would be

$$x_{prompt} = \text{[CLS]} \ x \ \text{Yeh [MASK] hai. [SEP]}$$

In the literature, various templates are utilized for prompt-based classification tasks (Gao et al., 2021). However, we find that "Yeh [MASK] hai." template is performing better for Urdu and Roman Urdu. We do not mention the results for other templates due to lack of space in the paper.

The language model $\mathcal{L}$ is then responsible for determining whether it is more suitable to fill in the [MASK] position with "khubsurat" (beautiful) or "fazool" (useless) as depicted in Figure 1. This prompt-based methodology enables the model to autonomously complete prompts and make sentiment/classification predictions, enabling more efficient and accurate classification.

For standard fine-tuning, we use the token sequence

$$x_{fine-tune} = \text{[CLS]Yeh jaga bohat pyari hai.[SEP]}$$

To construct the training and development sets for each dataset, we select $K$ labeled examples

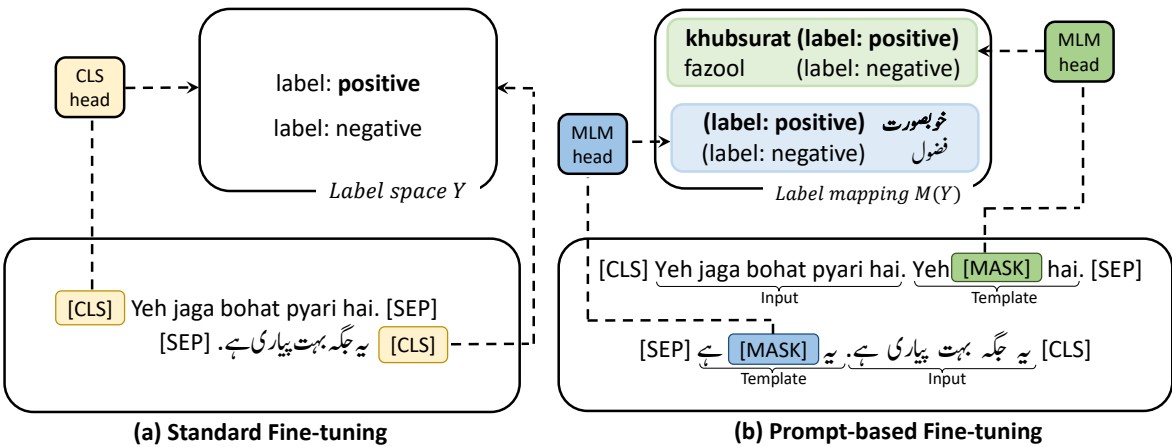

Figure 1: An illustration of (a) standard fine-tuning and (b) prompt-based fine-tuning for text classification in Urdu and Roman Urdu. The underlined text represents the task-specific template, designed explicitly for Urdu and Roman Urdu, while the bold words highlight the label words. The (Urdu/Roman Urdu text : and their English translation) is given as follows: (یہ جگہ بہت پیاری ہے : "This place is very beautiful."), (یہ [MASK] ہے : It is [MASK]), (Yeh [MASK] hai : It is [MASK]) (خوبصورت : beautiful), (فضول : useless)

from $D_{\text{train}}$ for each class, resulting in a total of $Y \times K$ labeled examples, where $Y$ represents the total number of classes (labels) in the dataset. In our experiments, we consider $K = 4, 8, 16$ and $32$. We also refer to this number as splits. The remaining examples from $D_{\text{train}}$ are reserved for the test set (with no labels). To ensure fair evaluation, we perform multiple rounds of testing. We select samples randomly from the unlabeled test set not used in each round's training $D_{\text{train}}$ and development $D_{\text{dev}}$ sets. This process is repeated five times, allowing for a comprehensive evaluation of the models' performance across different test sets.

We evaluate classification performance using accuracy and macro F1-score. We report the mean and standard deviation of each measure over the five runs and for different splits. This approach allows us to draw robust conclusions regarding the effectiveness of prompt-based and standard fine-tuning approaches for text classification.

Prompt-based fine-tuning involves using language-specific prompts tailored to Urdu and Roman Urdu to provide additional context to the models during training. This helps capture the intricacies of the languages and improve text classification performance. No prompts are used in standard fine-tuning, and the models rely solely on labeled examples. Comparing these approaches allows us to assess the impact of prompts in leveraging language-specific knowledge.

We used the Hugging Face[1] library for standard fine-tuning of models, while for prompt-based fine-tuning, we employed the OpenPrompt[2] library. OpenPrompt is an open-source framework designed explicitly for prompt learning, providing a comprehensive set of tools and resources for this approach. Our fine-tuning process (standard and prompt-based) utilizes a learning rate of $2e-5$. The optimization method is AdamW (an Adam optimizer variant), and the loss function is Cross-Entropy Loss. The number of epochs for each training is 10. Our study utilizes three pre-trained multilingual transformers, namely (1) xlm-roberta-base, (2) bert-base-multilingual-cased, and (3) distilbert-base-multilingual-cased, which are publicly available on HuggingFace library.

### 3.1 Datasets

Our study utilizes five distinct datasets that span different domains, including emotion and offensive language detection. Specifically, the Urdu Nastalique Emotions Dataset (UNED) (Bashir et al., 2023) consists of 1119 instances for emotion detection, featuring labels such as Neutral, Happy, Sad, Anger, Fear, and Love. The URDU OFFENSIVE DATASET (UOD) (Akhter et al., 2020) and Roman Urdu Dataset (RUD) (Akhter et al., 2020) play pivotal roles in offensive language detection, with instance counts of 2106 and 147116, respectively. These datasets employ Of-

---

[1] https://huggingface.co/
[2] https://thunlp.github.io/OpenPrompt/

fensive and Non-Offensive labels. The Roman Urdu Emotion Detection Dataset (RUED) (Arshad et al., 2019), consisting of 2961 instances, facilitates emotion detection with labels including Anger, Sad, Happy, and Neutral. The Roman Urdu Hate-Speech and Offensive Language Detection (RUHSOLD) (Rizwan et al., 2020) dataset comprising 10012 instances revolves around hate speech and offensive language detection, utilizing labels such as Hate Speech / Offensive Language and Non-Offensive. RUHSOLD dataset originally used six labels Abusive/Offensive, Sexism, Religious Hate, Profane, and Normal. However, to make the tasks easier, we converted this multiclass problem to a binary class problem. The datasets selected for our experiments, however, represent both binary and multiclass text classification problems.

## 4 Results and Discussion

Table 1 shows the mean classification accuracy for prompt-based and standard fine-tuned pre-trained language models on different datasets. Mean accuracy is reported for splits of 4, 8, 16, and 32. In the UNED dataset, XLM-RoBERTa demonstrated the highest accuracy (after prompt-based fine-tuning) of 44.8% on 16 splits, outperforming all other models. Similarly, for RUD dataset, fine-tuned XLM-RoBERTa remained the top performer with an accuracy of 84.8%. For the RUHSOLD dataset, fine-tuned XLM-RoBERTa consistently dominated in performance across all splits, achieving the highest accuracy of 63.8% on 32 splits.

On the RUED dataset, prompt-based fine-tuned XLM-RoBERTa outperformed other models with the highest accuracy of 35.6% for splits 32. However, for splits 8 and 16, prompt fine-tuned Distil-BERT and BERT-Multilingual models performed better, respectively. In the case of the UOD dataset, the results were more varied. Prompt-based fine-tuned XLM-RoBERTa outperformed other models for splits 4 and 16. However, for splits 8 and 32, prompt fine-tuned BERT-Multilingual yielded better results compared to the other models.

In our experiments, we consistently observed that prompt fine-tuned models outperformed standard fine-tuned models achieving up to 13% absolute improvement and 5.44% average improvement in accuracy across all tasks. Additionally, it is noteworthy that as the data splits increased, the models' accuracy also increased, as evident from Figures 2, 3, 4, 5, and 6. These figures show the

variation of accuracy with split size for UOD, RUD, RUED, RUHSOLD, and UNED datasets, respectively, under prompt-based and standard fine-tuning of the respective best-performing models. In general, accuracy improves with an increase in split size. However, for prompt-based fine-tuned models, the increase in accuracy was more significant compared to standard fine-tuned models. It is also worth noting that standard fine-tuning almost always lags behind prompt-based fine-tuning for all models and datasets for up to 32 splits, confirming that for the limited number of labeled examples, prompt-based fine-tuning is preferable.

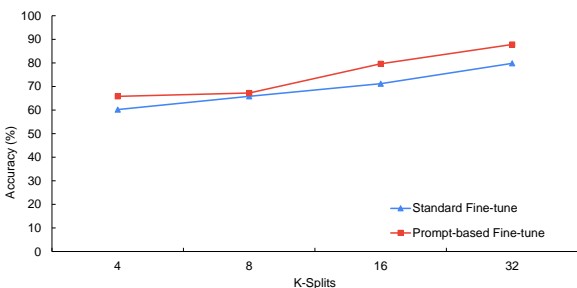

Figure 2: Accuracy comparison for Standard vs. Prompt-based Fine-tuning for UOD, where $K$-Split denotes the number of instances per class.

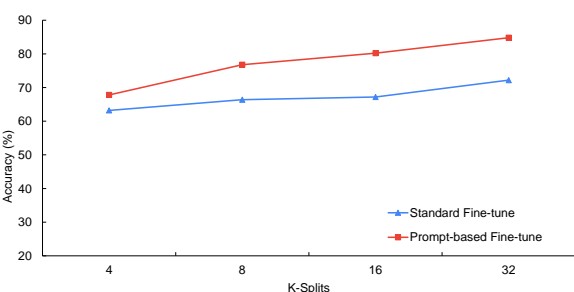

Figure 3: Accuracy comparison for Standard vs. Prompt-based Fine-tuning for RUD dataset, where $K$-Split denotes the number of instances per class.

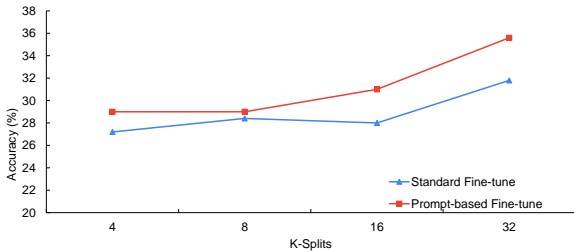

Figure 4: Accuracy comparison for Standard vs. Prompt-based Fine-tuning for RUED dataset, where $K$-Split denotes the number of instances per class.

| | | UNED | | | UOD | | | | RUD | | | | RUHSOLD | | | | RUED | | | |
|---|---|---|---|---|---|---|---|---|---|---|---|---|---|---|---|---|---|---|---|---|
| | | 4 | 8 | 16 | 4 | 8 | 16 | 32 | 4 | 8 | 16 | 32 | 4 | 8 | 16 | 32 | 4 | 8 | 16 | 32 |
| Standard fine-tuning | BERT-M | 21.4 | 26.8 | 32.0 | 60.2 | 65.8 | 71.2 | 79.8 | 63.2 | 66.4 | 67.2 | 72.2 | 53.0 | 55.0 | 57.6 | 60.0 | 27.2 | 28.4 | 28.0 | 31.8 |
| | DistilBERT | 18.8 | 20.8 | 23.4 | 58.6 | 57.4 | 68.4 | 80.0 | 55.0 | 64.2 | 63.2 | 71.0 | 49.2 | 55.6 | 56.6 | 60.4 | 27.4 | 29.8 | 31.6 | 25.4 |
| | XLM-R | 19.0 | 27.4 | 32.0 | 54.4 | 49.0 | 74.6 | 81.2 | 55.0 | 59.8 | 66.6 | 75.2 | 54.2 | 56.2 | 57.8 | 60.2 | 19.4 | 23.0 | 26.8 | 26.6 |
| Prompt based fine-tuning | BERT-M | 23.2 | 27.4 | 36.2 | 60.4 | **68.2** | 77.4 | **89.2** | 63.8 | 70.6 | 70.6 | 77.6 | 54.2 | 53.8 | 58.4 | 61.8 | 26.8 | 29.0 | **32.2** | 30.4 |
| | DistilBERT | 24.4 | 25.6 | 32.8 | 57.6 | 64.2 | 72.6 | 81.4 | 65.2 | 70.8 | 68.4 | 75.6 | 52.8 | 54.8 | 58.4 | 60.8 | 27.6 | **30.0** | 31.8 | 33.4 |
| | XLM-R | **27.4** | **31.8** | **44.8** | **65.8** | 67.2 | **79.6** | 87.8 | **67.8** | **76.8** | **80.2** | **84.8** | **57.2** | **59.0** | **64.0** | 63.8 | 29.0 | 28.4 | 31.0 | **35.6** |

Table 1: Mean accuracy comparison of standard and prompt-based fine-tuning for Urdu and Roman Urdu datasets using $K = 4, 8, 16$ and $32$, where $K$ represents the number of samples per class. The standard deviation of these experiments is given in Appendix A.

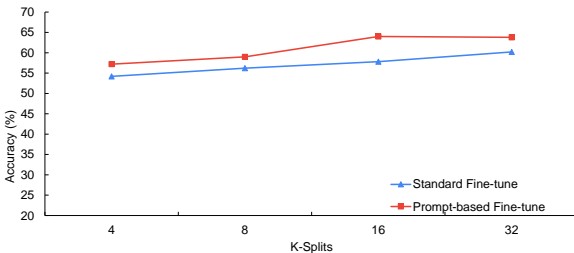

Figure 5: Accuracy comparison for Standard= vs. Prompt-based Fine-tuning for RUHSOLD dataset, where $K$-Split denotes the number of instances per class.

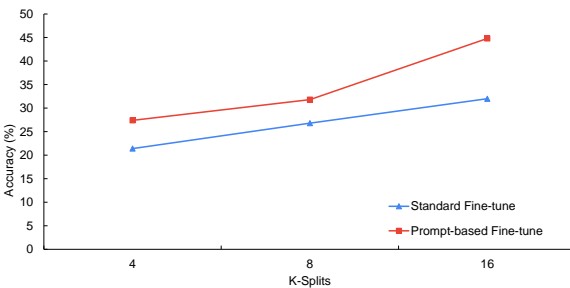

Figure 6: Accuracy comparison for Standard vs. Prompt-based Fine-tuning for UNED dataset, where $K$-Split denotes the number of instances per class.

We present additional experimental results in the Appendices. Appendix A gives the standard deviations corresponding to the mean accuracy values reported in Table 1. Appendix B presents the mean F1-score of all the experiments on different models and datasets. Appendix C studies the impact of fine-tuning by comparing the performance of zero-shot (no fine-tuning) and 4-shot prompt-based and standard fine-tuned models.

## 5 Conclusion

In this paper, we compare prompt-based fine-tuning with standard fine-tuning for Urdu and Roman Urdu text classification when restricted to dozens of training examples only. We perform experiments using several multilingual pre-trained language models on different classification datasets. Regardless of the training set size or the specific classification task, prompt-based fine-tuning consistently outperforms standard fine-tuning, highlighting its robustness and generalizability. The clear prediction superiority of prompt-based approaches, coupled with its generally lower computational cost, makes it an attractive alternative to traditional fine-tuning methods for low-resource languages. The insights gained from this study can inspire future research and encourage the adoption of prompt-based techniques in other low-resource languages.

## Limitations

It is essential to note that the presented experiments and results are relevant to Urdu and Roman Urdu. Consequently, the generalizability of the findings to other languages remains to be determined. The effectiveness and performance of the approach in diverse linguistic contexts may vary significantly. Thus, care should be exercised when inferring the results to languages beyond Urdu and Roman Urdu. Another significant factor to consider is the reliance of prompt-based techniques on domain expertise. The success of prompt-based fine-tuning heavily hinges upon formulating appropriate prompts that adequately capture the desired semantic information. Effective prompts require a deep understanding of the language, context, and the specific task. However, this process can introduce subjectivity and potential bias, as prompt design involves making subjective decisions and assumptions. These subjective elements may influence the performance of the approach and limit its objectivity in specific scenarios.

The current findings provide valuable insights into the utility of prompt-based fine-tuning for low-resource languages and text classification. Future

studies should investigate the performance of this approach across a broader range of languages and tasks, considering different linguistic characteristics and data availability. It is also essential to acknowledge that the evaluation of the proposed approach was focused solely on text classification. The applicability and performance of prompt-based fine-tuning in other NLP tasks, such as named entity recognition, sentiment analysis, or machine translation, still need to be explored. Therefore, caution should be exercised when attempting to generalize the findings to other NLP domains, as the effectiveness of prompt-based fine-tuning may vary depending on the task and its linguistic properties.

The limitations outlined in this study highlight the need for research and improvement in prompt-based fine-tuning. While the approach shows promise for low-resource languages and text classification, its generalizability, subjectivity in a prompt design, limited task scope, and broader data scarcity challenges necessitate further investigation and refinement. Addressing these limitations will enhance the applicability and effectiveness of prompt-based fine-tuning in diverse language settings and NLP tasks.

## Ethics Statement

We have carefully considered our study's ethical implications and taken the measures into account. Data privacy and confidentiality were strictly maintained throughout the research process. We consciously tried to mitigate biases and subjectivity in prompt design and analysis. Our approach is designed to assist human decision-making rather than replace it, emphasizing the importance of human involvement. We have given due consideration to the ethical aspects of prompt-based fine-tuning, including fairness and privacy issues. Our work contributes to advancing NLP knowledge and benefits various stakeholders. Overall, this study aligns with established ethical standards and promotes the responsible application of prompt-based fine-tuning in low-resource languages and text classification.

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

## A Standard Deviation of Accuracy

Table 2 shows the standard deviation of accuracy over 5 runs for different approaches, models, and datasets. The standard deviation corresponds to the mean accuracy reported in Table 1.

## B F1-score Comparison

Table 3 gives the mean F1-score for prompt-based and standard fine-tuning using different models and on different datasets. It is observed that F1-score results follow the same trend as that for accuracy results given in Table 1. Thus, despite significant class imbalance in some datasets the performance of text classification measured via accuracy and macro F1-score is fairly consistent.

| | | UNED | | | UOD | | | | RUD | | | | RUHSOLD | | | | RUED | | | |
|---|---|---|---|---|---|---|---|---|---|---|---|---|---|---|---|---|---|---|---|---|
| | | 4 | 8 | 16 | 4 | 8 | 16 | 32 | 4 | 8 | 16 | 32 | 4 | 8 | 16 | 32 | 4 | 8 | 16 | 32 |
| Standard fine-tuning | BERT-M | 6.06 | 3.96 | 7.0 | 4.20 | 2.28 | 2.38 | 3.56 | 3.42 | 0.89 | 3.11 | 2.38 | 3.46 | 4.63 | 5.02 | 2.82 | 4.60 | 4.66 | 5.04 | 2.78 |
| | DistilBERT | 3.27 | 3.96 | 4.27 | 1.94 | 4.72 | 6.42 | 2.0 | 4.84 | 2.48 | 3.34 | 2.23 | 2.04 | 1.81 | 1.14 | 1.51 | 6.58 | 5.67 | 5.17 | 3.28 |
| | XLM-R | 4.24 | 3.78 | 7.0 | 8.35 | 0.0 | 7.02 | 3.76 | 6.40 | 8.19 | 4.03 | 3.34 | 1.48 | 4.91 | 4.20 | 10.9 | 2.50 | 5.87 | 3.76 | 5.59 |
| Prompt based fine-tuning | BERT-M | 2.48 | 2.30 | 2.48 | 4.92 | 3.34 | 10.4 | 1.78 | 5.40 | 1.34 | 2.07 | 1.51 | 4.02 | 2.16 | 2.30 | 3.70 | 5.06 | 4.18 | 7.72 | 6.65 |
| | DistilBERT | 2.70 | 4.03 | 3.11 | 4.15 | 5.80 | 4.50 | 5.72 | 2.77 | 1.64 | 4.15 | 1.14 | 4.43 | 2.16 | 5.12 | 2.68 | 4.33 | 2.34 | 5.35 | 3.43 |
| | XLM-R | 2.19 | 3.03 | 7.01 | 9.47 | 6.05 | 5.72 | 2.77 | 5.35 | 2.77 | 4.20 | 2.94 | 3.76 | 3.08 | 4.52 | 3.70 | 4.24 | 9.39 | 2.0 | 6.76 |

Table 2: Standard deviation of classification accuracy across different datasets and pre-trained language models for Urdu and Roman Urdu datasets using $K = 4, 8, 16$ and $32$.

| | | UNED | | | UOD | | | | RUD | | | | RUHSOLD | | | | RUED | | | |
|---|---|---|---|---|---|---|---|---|---|---|---|---|---|---|---|---|---|---|---|---|
| | | 4 | 8 | 16 | 4 | 8 | 16 | 32 | 4 | 8 | 16 | 32 | 4 | 8 | 16 | 32 | 4 | 8 | 16 | 32 |
| Standard fine-tuning | BERT-M | 16.8 | 25.2 | 27.8 | 57.4 | 64.8 | 71.2 | 79.8 | 62.0 | 65.6 | 67.0 | 72.0 | 46.6 | 52.0 | 56.4 | 59.6 | 23.0 | 25.4 | 27.0 | 29.2 |
| | DistilBERT | 10.8 | 15.2 | 17.4 | 50.6 | 50.4 | 66.8 | 79.8 | 45.6 | 63 | 59.4 | 70.6 | 42.2 | 46.6 | 54.0 | 56.2 | 22 | 21.4 | 25.2 | 22.6 |
| | XLM-R | 10.0 | 23.8 | 27.8 | 43.4 | 33 | 73.2 | 80.8 | 45.2 | 52.6 | 62.8 | 75.0 | 46.0 | 40.8 | 56.8 | 56.4 | 12.4 | 15.2 | 22.4 | 23.4 |
| Prompt based fine-tuning | BERT-M | 20.4 | 23.4 | 32.8 | 59.2 | **67.6** | 76.0 | **89.2** | 63.4 | 69.8 | 70.0 | 77.4 | 52.6 | 51.4 | 54.4 | 61.8 | 22.8 | 25.0 | 28.0 | 26.8 |
| | DistilBERT | 20.4 | 22.8 | 29.4 | 55.2 | 63.4 | 72.4 | 81.4 | 64.2 | 70.6 | 67.8 | 75.6 | 51.8 | 53.2 | 57.6 | 60.6 | 24.6 | **27.2** | 27.8 | 30.0 |
| | XLM-R | **25.6** | **28.8** | **45.0** | **63.6** | 66.0 | **79.6** | 87.8 | **66.4** | **76.8** | **79.8** | **84.8** | **55.8** | **57.0** | **61.8** | **63.4** | **26.8** | 24.4 | **30.0** | **30.4** |

Table 3: Mean F1-score comparison of standard and prompt-based fine-tuning for Urdu and Roman Urdu datasets using $K = 4, 8, 16$ and $32$, where $K$ represents the number of samples per class.

| | | UNED | | UOD | | RUD | | RUHSOLD | | RUED | |
|---|---|---|---|---|---|---|---|---|---|---|---|
| | | Zero-Shot | 4-Shot | Zero-Shot | 4-Shot | Zero-Shot | 4-Shot | Zero-Shot | 4-Shot | Zero-Shot | 4-Shot |
| Standard | BERT-M | 9.0 | 21.4 | 50.0 | 60.2 | 50.0 | 63.2 | 54.0 | 53.0 | 21.0 | 27.2 |
| | DistilBERT | 18.0 | 18.8 | 51.0 | 58.6 | 51.0 | 55.0 | 54.0 | 49.2 | 16.0 | 27.4 |
| | XLM-R | 14.0 | 19.0 | 49.0 | 54.4 | 50.0 | 55.0 | 54.0 | 54.2 | 22.0 | 19.4 |
| Prompt based | BERT-M | 18.0 | 23.2 | 51.0 | 60.4 | 52.0 | 63.8 | 46.0 | 54.2 | 42.0 | 26.8 |
| | DistilBERT | 19.0 | 24.4 | 51.0 | 57.6 | 52.0 | 65.2 | 48.0 | 52.8 | 45.0 | 27.6 |
| | XLM-R | 16.0 | **27.4** | 51.0 | **65.8** | 50.0 | **67.8** | 48.0 | **57.2** | **46.0** | 29.0 |

Table 4: Accuracy results for zero-shot and 4-Shot for various models on different datasets.

| | | UNED | | UOD | | RUD | | RUHSOLD | | RUED | |
|---|---|---|---|---|---|---|---|---|---|---|---|
| | | Zero-Shot | 4-Shot | Zero-Shot | 4-Shot | Zero-Shot | 4-Shot | Zero-Shot | 4-Shot | Zero-Shot | 4-Shot |
| Standard | BERT-M | 5.0 | 16.8 | 34.0 | 57.4 | 33.0 | 62.0 | 35.0 | 46.6 | 17.0 | 23.0 |
| | DistilBERT | 5.0 | 10.8 | 34.0 | 50.6 | 40.0 | 45.6 | 40.0 | 42.2 | 9.0 | 22.0 |
| | XLM-R | 4.0 | 10.0 | 33.0 | 43.4 | 33.0 | 45.2 | 35.0 | 46.0 | 12.0 | 12.4 |
| Prompt based | BERT-M | 6.0 | 20.4 | 34.0 | 59.2 | 46.0 | 63.4 | 44.0 | 52.6 | 19.0 | 22.8 |
| | DistilBERT | **10.0** | 20.4 | 34.0 | 55.2 | 39.0 | 64.2 | 40.0 | 51.8 | 18.0 | 24.6 |
| | XLM-R | **10.0** | **25.6** | 35.0 | **63.6** | 33.0 | **66.4** | 38.0 | **55.8** | 16.0 | **26.8** |

Table 5: F1-score comparison for zero-shot and 4-Shot for various models on different datasets.

## C Accuracy and F1-Score Comparison for Zero-Shot and 4-Shot

Tables 4 and 5 present accuracy and F1-score, respectively, for zero-shot and fine-tuned with four examples on all five datasets. Results are shown for standard and prompt-based techniques using 3 pre-trained language models. It is clear from the results that zero-shot prompt-based predictions lag behind those for 4-shot prompt-based fine-tuned predictions on all the datasets. In other words, a performance boost is obtained by employing prompt-based fine-tuning even with 4 training examples. This underscores our argument that prompt-based fine-tuning holds significant potential in resource-constrained scenarios and is particularly relevant in the case of the Urdu language.