# OpenReview forum: "Comparing Prompt-Based and Standard Fine-Tuning for Urdu Text Classification"
_EMNLP/2023/Conference — EMNLP 2023 Findings_

### Official Review · Reviewer_pxrY · 2023-08-03

**Typos Grammar Style And Presentation Improvements:** 1. In Line 160, the word “and” betwee…
**Soundness:** 4

**Excitement:**

2: Mediocre: This paper makes marginal contributions (vs non-contemporaneous work), so I would rather not see it in the conference.

**Paper Topic And Main Contributions:**

This paper tackles Urdu text classification and makes a comparison between a prompt-based and standard fine-tuning method on Urdu text classification in a few-shot setting (K=4, 8, 16, and 32) using five datasets. These datasets contain both Urdu and Roman Urdu characters and consist of emotion detection and offensive language detection tasks. The paper verifies the superiority of the prompt-based method over the standard fine-tuning method with three pre-trained multilingual language models (i.e., BERT-Multilingual, DistilBERT, XLM-RoBERTa).

Updated: I have read the author rebuttal and adjusted the scores given its type.

**Questions For The Authors:**

A. Is there any special reason or motivation behind the choice of Urdu?

B. In Figure 1, what does the label mapping look like in each dataset used in the experiment? The prompt-based fine-tuning method requires defining the label mapping that maps target labels to words in the model’s vocabulary for each dataset. This mapping benefits subsequent research in this direction if available and should be considered as a contribution of the paper since the mapping used in the experiment has already been shown to work well in the paper.

C. How are the prompt-based few-shot results compared to those in a zero-shot setting? The prompt-based method can be evaluated in a zero-shot setting without any fine-tuning simply by taking a word with the highest probability among words in the label mapping at the [MASK] position. The comparison with the zero-shot results is needed to verify the effectiveness and necessity of fine-tuning in the prompt-based method.

D. When does the standard fine-tuning method outperform the prompt-based method as the number of samples per class, namely K, increases? In general, if a sufficient amount of labeled data is available, the standard fine-tuning method works quite well and could outperform the prompt-based method. Given that the datasets used in the paper have enough samples to test if the paper does not adopt the few-shot setting, it is recommended to analyze when the standard fine-tuning method outperforms the prompt-based method.

**Reasons To Accept:**

1. The paper studies the feasibility of a prompt-based fine-tuning method in Urdu, which is considered a low-resource language.
2. The paper experiments with three pre-trained language models and ensures that the outperformance of the prompt-based method over the standard fine-tuning method is consistent within these LMs.
3. The paper discusses its limitations from various aspects.

**Reasons To Reject:**

1. While it is important to address low-resource languages, it is unclear for the choice of Urdu language among many other low-resource languages such as Swahili and Nepali.
2. The main argument stated in the paper (in Lines 021-024 and 079-082) is somewhat overclaimed.
    - The paper states that prompt-based fine-tuning allows the models to capture Urdu’s unique linguistic characteristics and nuances, while it seems to lack the experiment and analysis to support this claim. It is necessary to qualitatively and/or quantitatively show what type of Urdu’s unique linguistic characteristics and nuances are captured by the model in addition to reporting the overall accuracy on the datasets. If it is difficult to fit additional experiments in the body of the short paper, it should be preferable to rewrite or tone down the argument.
3. A description of the experiment, especially for the fine-tuning procedure, is missing, which affects reproducibility.
    - For reproducibility, at least the following information should be provided: the source of the pre-trained language model and the hyper-parameters such as batch size, learning rate, number of epochs, optimization method, etc. As for the source of the pre-trained language model, there are multiple options, even for a single model. For example, BERT-Multilingual has two variants provided by Hugging Face (`bert-base-multilingual-uncased` and `bert-base-multilingual-cased`) while XLM-R offers two different sizes (`xlm-roberta-base` with 279M parameters and `xlm-roberta-large` with 561M parameters). Different variants of models could produce different results.

**Reproducibility:**

3: Could reproduce the results with some difficulty. The settings of parameters are underspecified or subjectively determined; the training/evaluation data are not widely available.

**Reviewer Confidence:**

4: Quite sure. I tried to check the important points carefully. It's unlikely, though conceivable, that I missed something that should affect my ratings.

---

> ### Author Rebuttal · Authors · 2023-08-28
>
> Point 1: While it is important to address low-resource languages, it is unclear for the choice of Urdu language among many other low-resource languages such as Swahili and Nepali.
>
> Response 1: Please see Response 4 below.
>
>
> Point 2: The main argument stated in the paper (in Lines 021-024 and 079-082) is somewhat overclaimed. The paper states that prompt-based fine-tuning allows the models to capture Urdu’s unique linguistic characteristics and nuances, while it seems to lack the experiment and analysis to support this claim. It is necessary to qualitatively and/or quantitatively show what type of Urdu’s unique linguistic characteristics and nuances are captured by the model in addition to reporting the overall accuracy of the datasets. If it is difficult to fit additional experiments in the body of the short paper, it would be preferable to rewrite or tone down the argument.
>
> Response 2: Thank you for your feedback. We will revise the statement to accurately represent our findings and their implications for prompt-based methods for Urdu and Roman Urdu. The statement would be updated as: "The improved classification accuracy achieved by prompt-based fine-tuning over standard fine-tuning suggests its potential as a valuable approach for low-resource languages with limited labelled data."
>
>
>
> Point 3: A description of the experiment, especially for the fine-tuning procedure, is missing, which affects reproducibility. For reproducibility, at least the following information should be provided: the source of the pre-trained language model and the hyper-parameters such as batch size, learning rate, number of epochs, optimization method, etc. As for the source of the pre-trained language model, there are multiple options, even for a single model. For example, BERT-Multilingual has two variants provided by Hugging Face (bert-base-multilingual-uncased and bert-base-multilingual-cased) while XLM-R offers two different sizes (xlm-roberta-base with 279M parameters and xlm-roberta-large with 561M parameters). Different variants of models could produce different results.
>
> Response 3: We appreciate your valuable feedback regarding the need for clarity in our experimental procedures. In response, we provide the missing information: our fine-tuning process utilizes a learning rate of 2e-5. The optimization method is AdamW (an Adam optimizer variant), and the loss function is Cross-Entropy Loss. The number of epochs for each training is 10. Our study utilizes three pre-trained multilingual transformers, namely (1) xlm-roberta-base, (2) bert-base-multilingual-cased, and (3) distilbert-base-multilingual-cased, which are publicly available on HuggingFace library.
>
> We provided the link to the HugginFace library in the manuscript; however, your suggestion to add additional details will improve the paper’s reproducibility.
>
>
>
> Questions For the Authors:
>
> Point 4: Is there any special reason or motivation behind the choice of Urdu?
>
> Response 4: We appreciate the reviewer's question about the choice of Urdu over other languages like Swahili and Nepali. The decision to focus on Urdu is primarily motivated by our deep familiarity and native-level proficiency in Urdu. This familiarity enables us to make well-informed decisions and conduct accurate analyses in Urdu, including the important issue of prompt design, which would not be the case with other languages like Swahili and Nepali. Additionally, we recognize Urdu's distinct right-to-left writing system and regional diversity. This distinctive feature brings specific challenges in terms of processing and analysis. While we acknowledge the importance of other low-resource languages, our familiarity with Urdu's linguistic intricacies and distinctive writing system makes it a logical choice for our research emphasis.
>
>
>
> Point 5: In Figure 1, what does the label mapping look like in each dataset used in the experiment? The prompt-based fine-tuning method requires defining the label mapping that maps target labels to words in the model’s vocabulary for each dataset. This mapping benefits subsequent research in this direction if available and should be considered as a contribution to the paper since the mapping used in the experiment has already been shown to work well in the paper.
>
> Response 5: We appreciate your valuable suggestion. In Figure 1, the label mapping for each dataset used in the experiment is not explicitly depicted due to the numerous datasets spanning different domains, such as sentiment analysis and hate speech, that are employed in this work. However, as you pointed out, we did indeed establish label mappings for the prompt-based fine-tuning method, and we will add the list of labels in the Appendix of the paper. These mappings were created by associating target labels with words from the model's vocabulary.
>
>
>
> Point 6: How are the prompt-based few-shot results compared to those in a zero-shot setting? The prompt-based method can be evaluated in a zero-shot setting without any fine-tuning simply by taking a word with the highest probability among words in the label mapping at the [MASK] position. Comparison with the zero-shot results is needed to verify the effectiveness and necessity of fine-tuning in the prompt-based method.
>
> Response 6: Thank you for the insightful comment. Our primary focus in the short paper is to evaluate fine-tuning with limited amounts of data. Obviously, zero-shot predictions will fare poorly as compared to few-shot fine-tuned predictions. Nonetheless, we will include zero-shot results in the final version of the paper.
>
>
>
> Point 7: When does the standard fine-tuning method outperform the prompt-based method as the number of samples per class, namely K, increases? In general, if a sufficient amount of labelled data is available, the standard fine-tuning method works quite well and could outperform the prompt-based method. Given that the datasets used in the paper have enough samples to test if the paper does not adopt the few-shot setting, it is recommended to analyze when the standard fine-tuning method outperforms the prompt-based method.
>
> Response 7: We appreciate your thoughtful feedback. We refrain from evaluating with large numbers of examples because we want to mimic the scenario where only a limited number of examples for a given task are available. This is quite common for low-resource languages. Furthermore, it is worth noting that prompt-based fine-tuning is often more economical than standard fine-tuning on many platforms.
>
> Please also see Response 2 to Reviewer 2.
>
>
>
> Point 8: Typos Grammar Style and Presentation Improvements: In Line 160, the word “and” between 16 and 32 should not be in italics, In Line 200, in 1 -> in Table 1, In Lines 213 and 215, The Table 1 -> Table 1
>
> Response 8: Thank you for pointing out these typos. We will fix them in the final version.
>
>
>
> Point 9: The brief description of the datasets used for the experiment (e.g., the number of examples in train/dev/test sets, the total number of target classes, and the characteristics of the texts, such as the source or domain of the texts) should be added to Appendix for the readers of the paper to better understand the tasks used for the evaluation.
>
> Response 9: Thank you for highlighting this aspect. We acknowledge the importance of providing datasets and experiment details. Our study utilizes five distinct datasets that span different domains, including emotion and offensive language detection. Specifically, the UNED dataset consists of 1119 instances for emotion detection, featuring labels such as Neutral, Happy, Sad, Anger, Fear, and Love. The UOD and RUD datasets play pivotal roles in offensive language detection, with instance counts of 2106 and 147116, respectively. These datasets employ Offensive and Non-Offensive labels. The RUED dataset, consisting of 2961 instances, facilitates emotion detection with labels including Anger, Sad, Happy, and Neutral. The RUHSOLD dataset, comprising 10012 instances, revolves around hate speech and offensive language detection, utilizing labels such as Hate Speech / Offensive Language and Non-Offensive. RUHSOLD dataset originally used six labels Abusive/Offensive, Sexism, Religious Hate, Profane, and Normal. However, to make the tasks easier, we converted this multiclass problem to a binary class problem.
>
> To generate robust evaluations, we employ a 5-fold cross-validation approach. This involved the creation of five samples for each dataset, using different splits (4, 8, 16, and 32) for training and development, whereas the remaining for testing set. The importance of non-repetition of training samples within training splits is diligently maintained.
>
> As recommended, we will include these details in the Appendix.

---

### Official Review · Reviewer_18m6 · 2023-08-03

**Soundness:** 3

**Excitement:**

2: Mediocre: This paper makes marginal contributions (vs non-contemporaneous work), so I would rather not see it in the conference.

**Missing References:**

Chandio et al. 2022, "Sentiment Analysis of Roman Urdu on E-Commerce Reviews Using Machine Learning" https://www.techscience.com/CMES/v131n3/47397
This paper handles Urdu sentiment analysis.  The task is three-way classification so it is harder than binary classification.

**Paper Topic And Main Contributions:**

This paper compares two approaches to Urdu binary sentiment classification: traditional supervised fine-tuning and prompt-based few-short learning.  The authors reported the results on five datasets written in Persian and Roman alphabets and three pre-trained language models.
Experimental results show that the prompt-based approach consistently outperformed the supervised method.

**Questions For The Authors:**

A: Please provide the corresponding English to the example sentences and templates (LL 148).

**Reasons To Accept:**

* Handled Urdu language that has not been well studied.
* Provided exhaustive experimental results for five data sets.

**Reasons To Reject:**

* Just reporting the results with known methods, without proposing new or better methods for prompting.
* The conclusion that prompt based tuning is better is not surprising at all, considering the standard fine-tuning was trained on very small data.  As done in Chandio 2022 (see missing references), normally the traditional fine-tuning requires at least thousands of data.
* The observation on the graphs (Fig 2 to 6), the more sample helps higher accuracy, is trivial.
* Five data sets are tested, but the overall differences between Persian and Roman alphabets are not discussed.  As the characteristics of Urdu, the difference of writing should be qualitatively and quantitatively compared.

**Reproducibility:**

3: Could reproduce the results with some difficulty. The settings of parameters are underspecified or subjectively determined; the training/evaluation data are not widely available.

**Reviewer Confidence:**

3: Pretty sure, but there's a chance I missed something. Although I have a good feel for this area in general, I did not carefully check the paper's details, e.g., the math, experimental design, or novelty.

**Typos Grammar Style And Presentation Improvements:**

* "as shown in 1" in LL200 should be "Table 1 & 2", I guess.
* "Ye" should be "Yeh" in Figure 1 (b)?
* LL231: "35.6% for splits 4 and 32" is not accurate - 35.6% is just for 32.

---

> ### Author Rebuttal · Authors · 2023-08-28
>
> Rebuttal Points:
>
> Point 1: Just reporting the results with known methods, without proposing new or better methods for prompting.
>
> Response 1: The primary focus of this short paper is to examine and contrast two distinct learning paradigms for Urdu text classification: prompt-based and Standard Fine-Tuning. The use of prompts is a relatively new technique that can be valuable for situations where training data is limited. Furthermore, there is little exploration into how these techniques function within the context of the Urdu language. To the best of our knowledge, there are no published works on prompt engineering or prompt-based fine-tuning for Urdu text classification. Our work comparing prompt and standard fine-tuning in Urdu can seed further research in this direction.
>
> Please see Response 2 to Reviewer 1 as well. Thanks!
>
> Point 2: The conclusion that prompt-based tuning is better is not surprising at all, considering the standard fine-tuning was trained on very small data. As done in Chandio 2022 (see missing references), normally the traditional fine-tuning requires at least thousands of data.
>
> Response 2: Our objective is to determine which modern-day learning paradigm performs better when only a limited amount of training data is available. This is a constraint common in low-resource language processing. Obviously, when large amounts of training data are available standard fine-tuning should be preferred.
>
> As highlighted by Reviewer 3 (Point 7), it would be useful to determine the number of training examples after which standard fine-tuning surpasses prompt-based fine-tuning. In our experiments, where training data is constrained to a few dozen examples, this did not occur.
>
> Point 3: The observation on the graphs (Fig 2 to 6), that more sample helps higher accuracy, is trivial.
>
> Response 3: We present these figures to emphasize the significance of the observation that increasing sample size improves text classification accuracy in the context of the low-resource language Urdu. In other words, this observation reinforces the fundamental understanding that larger sample sizes generally contribute to enhanced accuracy, but when we have scarce data, Prompt engineering can be better than standard techniques.
>
> Point 4: Five data sets are tested, but the overall differences between Persian and Roman alphabets are not discussed. As the characteristics of Urdu, the differences in writing should be qualitatively and quantitatively compared.
>
> Response 4: Thank you for your valuable feedback. The study of different scripts of Urdu and their implication on prompt-based text classification warrants separate treatment and is beyond the scope of the current short paper. Nonetheless, we will highlight this issue in the final version and suggest it as a future direction.
>
> Questions For the Authors:
>
> Point 5: Please provide the corresponding English to the example sentences and templates (LL 148).
>
> Response 5: Thank you for your suggestion. Please see Response 5 to Reviewer 1.
>
> Point 6: Missing References: Chandio et al. 2022, Sentiment Analysis of Roman Urdu on E-Commerce Reviews Using Machine Learning “https://www.techscience.com/CMES/v131n3/47397 This paper handles Urdu sentiment analysis.” The task is three-way classification, so it is harder than binary classification.
>
> Response 6: Thank you for your suggestion; we will add this reference to the final version of the paper.
> To clarify, our evaluation datasets include the task of sentiment classification and represent both binary and multiclass text classification tasks.
>
> Point 7: Typos Grammar Style and Presentation Improvements: "as shown in 1" in LL200 should be "Table 1 \& 2", I guess.
>
> Response 7: Thank you. You are right, we will correct this mistake in the final version of the paper.
>
> Point 8: "Ye" should be "Yeh" in Figure 1 (b)?
>
> Response 8: Thank you for the thorough review. This was a typo, and we will correct this in the final version of the paper.
>
> Point 9: LL231: "35.6\% for splits 4 and 32" is not accurate - 35.6\% is just for 32.
>
> Response 9: Thank you. You are right, we will correct this typo in the final version of the paper.

---

### Official Review · Reviewer_XVEi · 2023-08-05

**Typos Grammar Style And Presentation Improvements:** 1. Can you provide a parallel English…
**Soundness:** 3

**Excitement:**

2: Mediocre: This paper makes marginal contributions (vs non-contemporaneous work), so I would rather not see it in the conference.

**Paper Topic And Main Contributions:**

The paper compares the prompt-based and standard fine-tuning for Urdu text classification. More specifically, three pre-trained LMs have been evaluated on five Urdu text classification datasets. The experimental result shows that XLM-R model together with prompt-based tuning obtains the best classification performance.

**Questions For The Authors:**

1.  Can you provide the F1 result of Table 1?

**Reasons To Accept:**

The paper may be interesting to the Urdu language research community, considering that the Urdu language has less research and available resources than the English language.

**Reasons To Reject:**

1. The evaluation metric, accuracy, in Table 1 is not convincing. For example, the test set of Roman Urdu Hate-Speech and Offensive Language Detection (RUHSOLD) data [1] is unbalanced. In this case, compared with the F1 score, the accuracy may ignore the minor class.

2. Previous works have proposed the prompt-based approach, and the paper does not provide much insight in terms of methodology.

[1] https://aclanthology.org/2020.emnlp-main.197.pdf

**Reproducibility:**

3: Could reproduce the results with some difficulty. The settings of parameters are underspecified or subjectively determined; the training/evaluation data are not widely available.

**Reviewer Confidence:**

4: Quite sure. I tried to check the important points carefully. It's unlikely, though conceivable, that I missed something that should affect my ratings.

---

> ### Author Rebuttal · Authors · 2023-08-28
>
> Reasons to reject
> Point 1: The evaluation metric, accuracy, in Table 1 is not convincing. For example, the test set of Roman Urdu Hate-Speech and Offensive Language Detection (RUHSOLD) data [1] is unbalanced. In this case, compared with the F1-score, the accuracy may ignore the minor class.
>
> Response 1: We appreciate your comment regarding the class imbalance in RUHSOLD dataset, and the superiority of the F1-score in such scenarios. We have the F1-score results as well but decided to include the accuracy results in the short paper for two reasons: (1) the F1-score results show similar trends to accuracy results, and (2) most of the datasets we used have an even distribution of classes. Considering the short paper’s length limitations, we will provide the F1-score results in the Appendix. For your convenience, we are also presenting the F1-score table below. As you can see, a similar performance trend is observed in F1-score for all the datasets.
>
> |                          |            | UNED     |          |        | UOD      |          |          |          | RUD      |          |          |          | RUHSOLD  |        |          |          | RUED     |        |        |          |
> |--------------------------|------------|----------|----------|--------|----------|----------|----------|----------|----------|----------|----------|----------|----------|--------|----------|----------|----------|--------|--------|----------|
> |                          |            | 4        | 8        | 16     | 4        | 8        | 16       | 32       | 4        | 8        | 16       | 32       | 4        | 8      | 16       | 32       | 4        | 8      | 16     | 32       |
> | Standard fine-tuning     | BERT-M     | 16.8     | 25.2     | 27.8   | 57.4     | 64.8     | 71.2     | 79.8     | 62       | 65.6     | 67       | 72       | 46.6     | 52     | 56.4     | 59.6     | 23       | 25.4   | 27     | 29.2     |
> |                          | DistilBERT | 10.8     | 15.2     | 17.4   | 50.6     | 50.4     | 66.8     | 79.8     | 45.6     | 63       | 59.4     | 70.6     | 42.2     | 46.6   | 54       | 56.2     | 22       | 21.4   | 25.2   | 22.6     |
> |                          | XLM-R      | 10       | 23.8     | 27.8   | 43.4     | 33       | 73.2     | 80.8     | 45.2     | 52.6     | 62.8     | 75       | 46       | 40.8   | 56.8     | 56.4     | 12.4     | 15.2   | 22.4   | 23.4     |
> | Prompt based fine-tuning | BERT-M     | 20.4     | 23.4     | 32.8   | 59.2     | **67.6** | 76       | **89.2** | 63.4     | 69.8     | 70       | 77.4     | 52.6     | 51.4   | 54.4     | 61.8     | 22.8     | **25** | 28     | 26.8     |
> |                          | DistilBERT | 20.4     | 22.8     | 29.4   | 55.2     | 63.4     | 72.4     | 81.4     | 64.2     | 70.6     | 67.8     | 75.6     | 51.8     | 53.2   | 57.6     | 60.6     | 24.6     | 27.2   | 27.8   | 30       |
> |                          | XLM-R      | **25.6** | **28.8** | **45** | **63.6** | 66       | **79.6** | 87.8     | **66.4** | **76.8** | **79.8** | **84.8** | **55.8** | **57** | **61.8** | **63.4** | **26.8** | 24.4   | **30** | **30.4** |
>
>
> Point 2: Previous works have proposed the prompt-based approach, and the paper does not provide much insight in terms of methodology. [1] https://aclanthology.org/2020.emnlp-main.197.pdf
>
> Response 2: Our primary focus in the short paper is the comparison of two competing paradigms for utilizing large language models (LLMs). We were motivated by the fact that low-resource languages do not have large amounts of training data for different scenarios, and standard fine-tuning of LLMs can be computationally and financially more expensive than that for prompt-based fine-tuning of the same LLMs. As such, our main contribution is the demonstration that prompt-based fine-tuning can in fact outperform standard fine-tuning for different text classification tasks in a popular, yet under-resourced, language Urdu.
>
> We have utilized the RUHSOLD dataset introduced in Rizwan et al. (2020). They focus on hate speech and offensive language classification in Roman Urdu only, while we consider several text classification applications in Urdu and Roman Urdu. Our motivation and contribution, as highlighted above, is also different from theirs’.
>
> Questions For the Authors:
>
> Point 3: Can you provide the F1 result of Table 1?
>
> Response 3: Please see Response 1 above.
>
> Point 4: Typos Grammar Style and Presentation Improvements:
>
> Response 4: Your thorough review is much appreciated. We will correct these typos and grammatical mistakes in the final version of the paper.
>
> Point 5: Can you provide a parallel English translation for the Urdu text in Figure 1?
>
> Response 5: We will include the parallel English translation for the Urdu text featured in Figure 1 in its caption. The translation is also provided below for your reference.
>
> Urdu text: "Yeh jaga bohat pyari hai. yeh [MASK] hai."
>
> English: "This place is good. It is [MASK]."

---

### Meta-Review · Area_Chair_nwQV · 2023-09-20

**Recommendation:** 3

**Metareview:**

This paper compares prompt-based fine-tuning with standard fine-tuning for 5 text classification tasks from Urdu and Roman Urdu languages. These datasets cover emotion detection and offensive language detection tasks. The paper uses 3 pretrained LLMs and shows that prompt-based fine-tuning outperforms standard fine-tuning when using a few examples (4, 8, 16, 32 per class).

As reviewers mentioned, this work presents a valuable contribution focusing on Urdu as a low-resource language. During the discussion period, the authors highlighted that the presented comparison (prompt-based fine-tuning vs. standard fine-tuning) is important when fine-tuning with a very small amount of data which is in line with the paper's focus on Urdu language. In this context, experimental results are sound and consistent across 5 datasets and 3 LLMs.

Addressing feedback from the reviewers, the authors provided additional experimental results during the discussion period and acknowledged the suggestions from reviewers. Concretely, incorporating the additional results, rephrasing the main claim, adding further details on the dataset and fine-tuning hyperparameters will further improve the paper.

---

### Decision · Program_Chairs · 2023-10-07

**Decision:**

Accept-Findings

**Comment:**

This paper compares prompt-based fine-tuning with standard fine-tuning for 5 text classification tasks from Urdu and Roman Urdu languages. These datasets cover emotion detection and offensive language detection tasks. The paper uses 3 pretrained LLMs and shows that prompt-based fine-tuning outperforms standard fine-tuning when using a few examples (4, 8, 16, 32 per class).

As reviewers mentioned, this work presents a valuable contribution focusing on Urdu as a low-resource language. During the discussion period, the authors highlighted that the presented comparison (prompt-based fine-tuning vs. standard fine-tuning) is important when fine-tuning with a very small amount of data which is in line with the paper's focus on Urdu language. In this context, experimental results are sound and consistent across 5 datasets and 3 LLMs.

Addressing feedback from the reviewers, the authors provided additional experimental results during the discussion period and acknowledged the suggestions from reviewers. Concretely, incorporating the additional results, rephrasing the main claim, adding further details on the dataset and fine-tuning hyperparameters will further improve the paper.